# Predictors to Intensive Care Unit admission among patient with coronavirus disease in Sukraraj Tropical and Infectious Disease Hospital, Nepal: A case-control study

**Dipsikha Aryal**[1]*, **Paras Kumar Pokharel**[1], **Anup Ghimire**[1], **Vijay Kumar Khanal**[1], **Gyanu Nepal Gurung**[1], **Bimal Sharma Chalise**[2], **Sudikshya Neupane**[1], **Shikha Basnet**[1]

1 B.P Koirala Institute of Health Sciences, School of Public Health and Community Medicine, Dharan, Nepal,
2 Sukraraj Tropical and Infectious Disease Hospital, Teku, Nepal

* seekuary@gmail.com

**Data Availability Statement:** Data can be found in OSF: https://osf.io/az4wr/.

## Abstract

The clinical features of COVID-19 are vary widely, ranging from asymptomatic states or mild upper respiratory tract infections to severe pneumonia. Previous studies have shown that 20.0% of COVID-19 patients are hospitalized, out of which 10.0–20.0% are admitted to the Intensive Care Unit. The present study aims to assess predictors associated with COVID-19 leading to Intensive Care Unit admission among reverse transcriptase- polymerase chain reaction (RT-PCR) positive patients in Sukraraj Tropical and infectious disease hospital, Nepal. A case-control study was conducted from June 2022 to July 2022 among patients admitted to Sukraraj Tropical and Infectious Disease Hospital. A hospital-based age (± 2 years) and sex-matched case-control study design were adopted in which ICU admitted (case group, n = 33) and general ward admitted (control group, n = 66) were included. Data were collected using a structured questionnaire comprising of socio-demographic, clinical, and preventive predictors. Data were analyzed using the Statistical Package for Social Science version 11.5. The Chi-square test and conditional logistic regression to determine the predictors associated with ICU admission. High blood pressure, high C-reactive protein and poor application of preventive practices were found to be the predictors of ICU admission. Conditional logistics regression analyses revealed that independent risk factors associated with ICU admission were elevated blood pressure (AOR = 2.22; 95% CI 1.05–4.71, p = 0.015) and abnormal C-Reactive Protein (AOR = 2.92; 95% CI 1.24–6.84, p = 0.012) at the time of hospital admission were more likely to get admitted to ICU. Likewise, patients with poor preventive practice (AOR = 3.34; 95% CI 1.19–9.31, p = 0.02) more likely to get admitted to ICU than patient with good preventive practices.These research findings hold potential significance for facilitating early triage and risk assessment in COVID-19 patients.

**Funding:** The authors received no specific funding for this work.

**Competing interests:** The authors have declared that no competing interests exist.

## Introduction

Coronavirus disease 2019 (COVID-19) is caused by the severe acute respiratory syndrome coronavirus 2 (SARS-CoV-2), previously known as 2019-nCoV. It was initially identified in Wuhan, China, in December 2019 and declared a pandemic by World Health Organization in March 2020 [1].

The clinical manifestations of COVID-19 vary widely, ranging from asymptomatic or mild upper respiratory tract infections to severe pneumonia with respiratory failure, acute respiratory distress syndrome, or fatal outcomes. Previous studies have delineated several major symptoms of COVID-19, including fever, cough, and dyspnea, alongside minor symptoms such as anosmia, gastrointestinal disturbances, and cutaneous manifestations [2].

While the majority of patients experience a favorable clinical course, approximately 20% necessitate hospitalization, with a subset progressing rapidly to acute respiratory distress syndrome (ARDS), multiple organ dysfunction syndrome, or mortality [3]. Rates for Intensive Care Unit (ICU) admission remain relatively low, with 10–20% of cases requiring intensive care, 3–10% necessitating intubation, and 2–5% resulting in death. Severely affected patients commonly present with respiratory failure, pneumonia, multi-organ failure, and shock, necessitating management in ICU [4].

Nepal reported its first case of coronavirus on 9 January 2020, in an individual returning from Wuhan City, who subsequently sought care at Sukraraj Tropical and Infectious Disease Hospital on 13 January 2020. The National Public Health Laboratory (NPHL) confirmed the positive test result for 2019-nCoV on 23 January 2020 [5]. On 23rd March 2020, Nepal reported its second case of COVID-19, prompting the initiation of a nationwide lockdown on March 25, 2020 [6]. Projections estimated 111,300 expected cases from September 2021 to April 2022, among which 3,590 would require ICU care [7]. Studies show the basic reproductive number during the second wave was larger than the first wave [7].

Various containment and mitigation strategies have been implemented globally to manage COVID-19, aiming to mitigate hospital surges and safeguard vulnerable populations. These strategies encompass clinical management guidelines, protocols for personal protective equipment usage, and waste management guidelines, particularly targeting elderly individuals and those with comorbidities [8]. Nonetheless, challenges in implementing these preventive measures at the community level persist due to cultural diversity, socioeconomic disparities, and healthcare service inequities [9].

In Nepal, ongoing clinical studies on COVID-19 have been predominantly focused on describing epidemiological and clinical characteristics, often comparing survivor and non-survivor groups or RT-PCR positive and negative cohorts. However, data comparing ICU-admitted and non-admitted groups remain scarce. Comprehensive analyses examining demographic, comorbidity, and clinical, and public health predictors associated with ICU admission in this context are lacking. Thus, we conducted a study to assess predictors associated with COVID-19 leading to ICU admission among RT-PCR-positive patients for SARS-CoV-2 at Sukraraj Tropical and Infectious Disease Hospital.

## Materials and methods

### Study design, population and setting

We conducted a hospital-based matched case-control study, in which individuals were matched based on age (±2 years) and sex. The proposal writing commenced on April 1, 2022, with data collection occurring from June 24, 2022, to July 12, 2022. The study involved patients admitted to the COVID-19 ICU and general ward of Sukraraj Tropical and Infectious Disease

Hospital, who tested positive for SARS-COV-2 via reverse transcriptase polymerase chain reaction.

### Study site

According to the Ministry of Health and Population's most recent updates as of August 6, 2021, there are currently 78 government hospitals operating COVID-19 care units. Within the Kathmandu Valley, there exist 10 public hospitals equipped with High Dependency Units (HDU), Intensive Care Unit (ICU) and ventilators. Among these establishments, Sukraraj Tropical and Infectious Disease Hospital was selected as the focal point for this study as it stands as Nepal's sole central government infectious disease hospital, entirely dedicated to managing COVID-19 cases. Sukraraj Tropical and Infectious Disease Hospital is equipped with 26 general beds, 50 HDU beds, and 24 ICU beds complemented with 22 ventilators, which are utilized in conjunction with the ICU beds. Moreover, this hospital serves as a certified laboratory for Reverse Transcriptase-Polymerase Chain Reaction (RT-PCR) testing and offers comprehensive COVID-19 care alongside ICU facilities. It holds a reputation as the most reliable healthcare institution among the general public.

### Sample size

A sample of 33 cases and 66 control was chosen in (1:2 ratio), the total number of samples was 99.

### Sampling technique

Sukraraj Tropical and Infectious Disease Hospital was selected from the list of public hospitals equipped with HDU, IDU, and ventilators in Kathmandu Valley. A compilation of data concerning COVID-19 RT-PCR positive patients admitted to the ICU and general ward (designated as Covid ward or Covid isolation ward) in Sukraraj Tropical and Infectious Disease Hospital was collected from Shrawan to Poush 2078 BS (16th July 2021 – 14th January 2022). The complete frame of ICU-admitted patients, numbering 222 over six months, was classified based on the calculated sample size for this investigation, set at 99. Each 2.24 ~ 3rd ICU-admitted patient file was carefully examined, wherein the baseline characteristics, oxygen saturation, blood pressure, pulse rate, respiration rate, and temperature at the time of hospital admission, along with laboratory electronic records such as Total Leucocyte Count (TLC), C-Reactive Protein (CRP) and Cycle Threshold Value (CT value) were meticulously documented. Systematic random sampling was employed in case selection; if the chosen 3rd respondent failed to meet the inclusion criteria, then the subsequent was assessed.

For the selection of controls, COVID-19 RT-PCR-positive individuals with at least one or more symptoms during the clinical course of the disease, admitted to the general ward, were chosen, according to age and sex-matching with the cases. Data on preventive measures were acquired through direct communication with the patients via telephone.

Among the total of 713 COVID-19 patients admitted during the six months, 2.1% of the participants were deemed ineligible for the study due to age limitations. Of the remaining 698 patients, 222 were admitted to the ICU and 371 to the COVID-19 general ward. Among these, 105 (15.04%) were excluded due to mortality during treatment. Subsequently, out of the initial 593 patients, 143 files were reviewed, data were documented, and each participant was contacted. Among the 143 contacted, 2 (1.39%) were excluded due to post-discharge mortality. From the remaining 141 participants, 33 cases were successfully matched with 66 controls, with the remaining 42 unmatched participants being excluded from this study, explained in Fig 1.

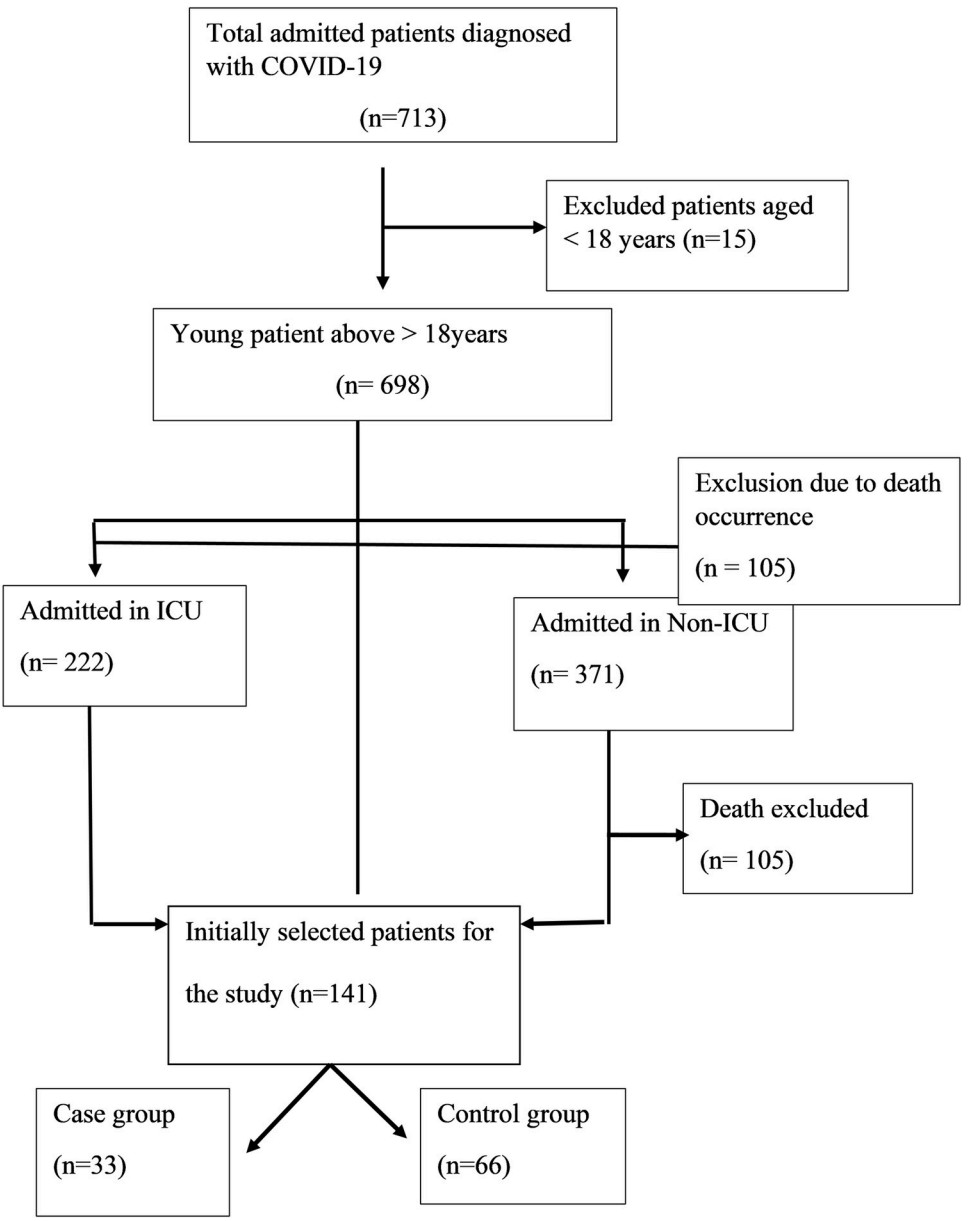

**Fig 1. Patient enrollment flowchart.**

## Populations / participants selection criteria

**Inclusion criteria for cases.** Patients eligible for inclusion were those who tested positive for SARS-CoV-2 via RT-PCR and were admitted to the COVID ICU of Sukraraj Tropical and Infectious Disease Hospital between July 16, 2021, and January 14, 2022. Included patients comprised of those who either fully recovered, recovered with sequelae, were referred to other facilities, or left against medical advice after ICU admission. Furthermore, only adults aged 18 years or older were considered for this study. For inquiries on preventive practices, individuals who provided verbal consent were incorporated.

**Inclusion criteria for control.** Participants qualified as controls if they tested positive for SARS-CoV-2 via RT-PCR and were admitted to the COVID ward, isolation ward, or cabin of

Sukraraj Tropical and Infectious Disease Hospital. This group encompassed patients who fully recovered, recovered with sequelae, were referred to other facilities, or were left against medical advice following admission. Additionally, controls experienced at least one subjective symptom during the clinical course of the disease. The eligibility criteria also specified adults aged 18 years and above. Those willing to provide verbal consent were included in inquiries concerning preventive practices

**Exclusion criteria for cases and control.** Exclusion criteria encompassed individuals who tested positive for SARS-CoV-2 via RT-PCR but were not admitted to Sukraraj Tropical and Infectious Disease Hospital. Deceased patients during or after admission to the ICU or COVID-19 ward, individuals under the age of 18 years, those with positive COVID-19 test results from rapid diagnostic tests, pregnant women, severely ill patients, and individuals with cognitive impairment—manifested by inability to identify the year, month, district, age, year of birth, or address—were all excluded from the study. Furthermore, those who did not provide verbal consent were not considered.

**Ethical approval.** Before conducting the study, ethical clearance was obtained from the Institutional Ethical Review Committee of B.P. Koirala Institute of Health Sciences (Reference number: 236/078/079-IRC). Additionally, study permission was secured from Sukraraj Tropical and Infectious Disease Hospital (Reference number: 078/79 444). Given the retrospective nature of the study, it was impractical to visit each respondent. Therefore, verbal consent was obtained from all participants before their enrollment. Throughout the study, the confidentiality and anonymity of the participants were rigorously maintained and ensured.

**Data collection.** Data was collected in a retrospective way using a structured questionnaire comprising three sections- a) Socio-demographics characteristics, b) Clinical predictors, c) Preventive predictors.

**Assessment of socio-demographic and clinical predictors.** Clinical data upon hospital admission, encompassing demographic particulars, chronic comorbidities, vital signs, symptoms, severity diagnosis of COVID-19, admission and discharge dates, COVID-19 vaccination status, and outcomes, were extracted from medical records utilizing a standardized data collection form. The severity at admission was categorized into four groups: mild, moderate, severe, and critical, based on the World Health Organization (WHO) COVID-19 Clinical Management Guidelines (Living Guidance 25 January 2021) [10, 11].

Lab parameters including the Threshold cycle (CT) value, Total Leucocyte Count (TLC) and C-reactive protein levels were retrived from an electronic records of a laboratory.

**Assessment of preventive predictors.** A telephonic interview was conducted with each case and control to ascertain the preventive measures adopted by the patient within the two weeks preceding COVID-19 infection. Additionally, to gather the socio-demographics, epidemiological, and symptom-related data not available in medical records, direct communication with patients or their families was established. The inquiry encompassed the following components:

Marital status, ethnicity, religion, occupation, family type, family size, means of transportation, smoking and alcohol consumption habits. For the assessment of preventive practice, the "Questionnaire to assess prevention practice against COVID-19 in the General Population" developed by AIIMS New Delhi, India was employed.

**Description of preventive practice questionnaire.** "Questionnaire to assess prevention practice against COVID-19 in the General Population" was developed by AIIMS New Delhi India [12]. (Please refer to Annexure–I and Annexure–II for the Questionnaire form in English and Nepali). The questionnaire comprises two sections: the first section evaluates preventive practices, while the second section explores reasons for non-adherence to preventive measures. Section A comprises 18 questions, wherein question 1 pertains to hand hygiene,

questions 2, 3, and 5 address social distancing, questions 6, 7, and 10 focus on mask usage, questions 4, 14, and 15 relate to basic precautions, questions 8, 9, and 16 concern adherence to safety guidelines and government restrictions, and questions 17 and 18 pertain to healthcare behavior. For this study, only Section A of the questionnaire was utilized.

**Statistical analysis.** Data were collected using a paper-based questionnaire and entered into the Statistical Package for Social Sciences (SPSS) version 11.5. Case-control matching was executed within SPSS. Parametric numerical variables were presented as mean and standard deviation, while categorical variables were expressed as percentages and frequencies. The normality of data distribution was assessed using the Kolmogorov–Smirnov and Shapiro–Wilk tests. The chi-square test was employed for categorical variables, and the Mann-Whitney U test was utilized to compare the association between numerical variables and ICU admission. Conditional logistic regression was conducted to ascertain predictors associated with ICU admission. The results of conditional logistic regression were presented using adjusted odds ratios (AORs), along with their 95% confidence intervals and p-values. All tests were conducted at a 95% confidence level, and p-values less than 0.05 were deemed statistically significant.

## Results

A total of 33 patients admitted to the ICU and 66 patients admitted in general ward were selected from among 713 patients hospitalized during the study period at participating centers. Table 1 shows the median age of cases was 59 (42–70) years, while the median age of control was 59 (43.25–70.00) years. Among married, 35.7% were cases and 64.3% of control. Analyzing factors such as current residence in municipality (38.5% vs 61.5%, p = 0.255), secondary education (37.5% vs 62.5%, p = 0.104), unemployed (34.1% vs 65.9%, p = 0.138).

Within the Brahmin/ Chhetri, 31.0% constituted case and 69.0% were control. Moreover, Hindu accounts for 34.1% cases and 65.9% control. Among those from nuclear family, 40.0% were case and 60.0% control. However, statistical analysis did not reveal significant associations between these socio-demographic variables and ICU admission.

Table 2 present the association between co-morbidities and behavioural characteristics with ICU admission. Pre-existing co -morbidities including diabetes, hypertension, cardiac disease, COPD and other medical issues were not significantly associated with ICU admission. Among behavioural characteristics, smoking was found to be significantly associated (p = 0.005) with ICU admission. Smoker was 3.4 (95% CI: 1.4–8.3) times likely admitted in ICU than those who were non-smoker. The odds of ICU admission were 20.0 (95% CI: 5.4–72.83; p value: <0.001) times higher in patient with low oxygen saturation than normal oxygen. Patients with elevated blood pressure had 6.10 (95% CI: 2.40–15.50; p value: <0.001) odds of ICU admission as compared to normal blood pressure. Patients with pulse rate above 100 bpm had 7.75 (95% CI: 2.23–26.87; p value: <0.001) times higher odds of ICU admission as compared to normal pulse rate. Febrile patient had 14.44 (95% CI: 1.65–125.75; p value: 0.005) chances of ICU admission in comparison with afebrile patients. Patients with tachypnea had 15.5 (95% CI: 3.4–70.10; p value: <0.001) higher odds of ICU admission in comparison with normal respiration rate.

Patients with abnormal Total Leucocyte Count had 10.07 (95% CI: 2.94–34.4; p value: <0.001) more chance of ICU admission in contrast with normal TLC. Patients with abnormal CRP had 7.18 (95% CI: 2.77–18.65; p value: <0.001) chances of ICU admission in comparison with normal CRP.

Patients without COVID-19 vaccination had 6.78 (95% CI: 2.63–17.48; p value: <0.001) higher chance of ICU admission in contrast to those vaccinated either first or second dose with any kind of vaccine.

**Table 1. Association between sociodemographic characteristics with ICU admission.**

| Variables | ICU admission (Cases) n (%) | General admission (Control) n (%) | p- value |
|---|---|---|---|
| **Age** | 59 (42–70) | 59 (43.25–70) | 0.970 |
| **Age group** | | | |
| 18–39 years | 6 (33.3) | 12 (66.7) | 0.084[a] |
| 40–59 years | 11 (33.3) | 22 (66.7) | |
| 60–79 years | 14 (34.1) | 27 (65.9) | |
| 80 + | 2 (28.6) | 5 (71.4) | |
| **Gender** | | | |
| Male | 22 (33.3) | 44 (66.7) | 1.000 [a] |
| Female | 11 (33.3) | 22 (66.7) | |
| **Marital Status** | | | |
| Married | 30 (35.7) | 54 (64.3) | 0.373 [b] |
| Others (Unmarried, Widow) | 3 (20.0) | 12 (80.0) | |
| **Permanent Address** | | | |
| Bagmati Province | 22 (33.3) | 44 (66.7) | 1.000 [a] |
| Others (Lumbini, Gandaki) | 11 (33.3) | 22 (66.7) | |
| **Current residence** | | | |
| Municipality | 20 (38.5) | 32 (61.5) | 0.255 [a] |
| Others (Metropolitan, Sub metropolitan city) | 13 (27.7) | 34 (72.3) | |
| **Educational Level** | | | |
| Illiterate | 3 (15.8) | 16 (84.2) | 0.104 [b] |
| Others (Secondary, higher) | 30 (37.5) | 50 (62.5) | |
| **Occupation** | | | |
| Home maker | 8 (47.1) | 9 (52.9) | 0.138 [c] |
| Unemployed | 15 (34.1) | 29 (65.9) | |
| Employed | 10 (26.3) | 28 (73.7) | |
| **Health Professionals** | | | |
| Yes | 0 (0.0) | 4 (100.0) | 0.298 [b] |
| No | 33 (34.7) | 62 (65.3) | |
| **Ethnicity** | | | |
| Brahmin/Chhetri | 18 (31.0) | 40 (69.0) | 0.564 [a] |
| Others (Dalit, Newar) | 15 (36.6) | 26 (63.4) | |
| **Religion** | | | |
| Hindu | 31 (34.1%) | 60 (65.9) | 0.715 [b] |
| Others (Buddhist, Christian) | 2 (25.0) | 6 (75.0) | |
| **Family type** | | | |
| Nuclear | 22 (40.0) | 33 (60.0) | 0.137 [a] |
| Others (Joint, Extended) | 11 (25.0) | 33 (75.0) | |

[a] = Pearson chi square

[b] = Fischer's exact test

[c] = Linear by linear chi square

Table 3. illustrates the severity of COVID-19 at the time of hospital admission. The variables presented in the table are significantly associated with ICU admission.

Table 4 displays the results of conditional logistics regression analysis. Patients exhibiting elevated blood pressure (adjusted odds ratio [AOR] = 2.22; 95% confidence interval [CI] 1.05–4.71; p = 0.015) and abnormal levels of C-reactive protein (AOR = 2.92; 95% CI 1.24–6.84;

**Table 2. Association between different predictors with ICU admission.**

| Variables | ICU admission (Cases) n (%) | General admission (Control) n (%) | χ2 test p value | Crude odd's ratio (95% CI) | |
|---|---|---|---|---|---|
| **Pre-existing Co-morbid illness** | | | | | |
| No (Ref) | 14 (25.9) | 40 (74.1) | 0.087 [a] | 2.08 (0.89 – 4.8) | |
| Yes | 19 (42.2) | 26 (57.8) | | | |
| **Diabetes Mellitus** | | | | | |
| No (Ref) | 28 (32.6) | 58 (67.4) | 0.674 [a] | 0.77 (0.23 – 2.57) | |
| Yes | 5 (38.5) | 8 (61.5) | | | |
| **Hypertension** | | | | | |
| No (Ref) | 22 (31.4) | 48 (68.6) | 0.532 [a] | 0.75 (0.30 – 1.85) | |
| Yes | 11 (37.9) | 18 (62.1) | | | |
| **Cardiac disease** | | | | | |
| No (Ref) | 31 (32.3) | 65 (67.7) | 0.257 [b] | 0.23 (0.02 – 2.7) | |
| Yes | 2 (66.7) | 1 (33.3) | | | |
| **COPD** | | | | | |
| No (Ref) | 26 (30.2) | 60 (69.8) | 0.092 [a] | 0.37 (0.11 – 1.21) | |
| Yes | 7 (53.8) | 6 (46.2) | | | |
| **Other medical issues** | | | | | |
| No (Ref) | 31 (33.7) | 61 (66.3) | 0.782 [b] | 0.78 (0.14 – 4.29) | |
| Yes | 2 (28.6) | 5 (71.4) | | | |
| **Smoking** | | | | | |
| Non- smoker (Ref) | 15 (23.4) | 49 (76.6) | **0.005** [a] | 3.4 (1.4 – 8.3) | |
| Smoker | 18 (51.4) | 17 (48.6) | | | |
| **Alcohol** | | | | | |
| Non-Alcoholic (Ref) | 17 (29.8) | 40 (70.2) | 0.388 [a] | 1.4 (0.62 – 3.36) | |
| Alcoholic | 16 (38.1) | 26 (61.9) | | | |
| **Oxygen saturation** | | | | | |
| Normal (Ref) | 3 (6.4) | 44 (93.6) | **<0.001*** | 20.00 | 5.4 - 72.83 |
| Below | 30 (57.7) | 22 (42.3) | | | |
| **Blood Pressure** | | | | | |
| Normal (Ref) | 14 (20.6) | 54 (79.4) | **<0.001*** | 6.10 | 2.40 – 15.50 |
| Elevated | 19 (61.3) | 12 (38.7) | | | |
| **Pulse rate** | | | | | |
| Normal (Ref) | 22 (26.2) | 62 (73.8) | **<0.001*** | 7.75 | 2.23 - 26.87 |
| Above | 11 (73.3) | 4 (26.7) | | | |
| **Temperature** | | | | | |
| Afebrile (Ref) | 27 (29.3) | 65 (70.7) | **0.005*** | 14.44 | 1.65 - 125.75 |
| Febrile | 6 (85.7) | 1 (14.3) | | | |
| **Respiration rate** | | | | | |
| Normal (Ref) | 2 (5.7) | 33 (94.3) | **<0.001*** | 15.5 | 3.4 – 70.10 |
| Tachypnea | 31 (48.4) | 33 (51.6) | | | |
| **Total leucocyte count** | | | | | |
| Normal (Ref) | 20 (24.4) | 62 (75.6) | **<0.001*** | 10.07 | 2.94 – 34.4 |
| Abnormal | 13 (76.5) | 4 (23.5) | | | |
| **C-reactive protein** | | | | | |
| Normal (Ref) | 8 (14.8) | 46 (85.2) | **<0.001*** | 7.18 | 2.77 – 18.65 |
| Abnormal | 25 (55.6) | 20 (44.4) | | | |
| **COVID-19 vaccination** | | | | | |

(*Continued*)

**Table 2.** (Continued)

| Variables | ICU admission (Cases) n (%) | General admission (Control) n (%) | χ2 test p value | Crude odd's ratio (95% CI) | |
|---|---|---|---|---|---|
| Yes (Ref) | 14 (20.3) | 55 (79.7) | <**0.001**\* | 6.78 | 2.63 – 17.48 |
| No | 19 (63.3) | 11 (36.7) | | | |

[a] = Pearson chi- square

[b] = Fischer's exact test

p = 0.012) upon hospital admission manifested a heightened likelihood of ICU admission. Additionally, patients demonstrating poor adherence to preventive measures (AOR = 3.34; 95% CI 1.19–9.31; p = 0.02) exhibited a higher odds for ICU admission compared to those with good adherence to preventive practices.

## Discussion

The current investigation revealed that the proportion of ICU admissions among COVID-19-infected patients was 31.13%, a figure consistent with a study conducted in Wuhan, where 32% of COVID-19-infected patients required ICU admission due to higher need for oxygen [2]. Among the data collected from 20 regions, the average ICU admission rate was 21.4%, with a range spanning from 9.4% to 45.9% [13]. Moreover, another study highlights that 20% of hospitalized patients with COVID-19 experienced severe symptoms that required intensive care [14]. Furthermore, a meta-analysis reported a pooled rate of ICU admission to be 32% in COVID-19 patients [15]. However, a slightly lower proportion of ICU admission was observed in a study conducted in South Korea, which reported that a total of 9.5% of COVID-19-infected patients required ICU care, with only 4.6% of patients being admitted to ICU [16].

The present study did not find any association between co-morbidities like diabetes mellitus, hypertension, cardiac disease, COPD with ICU admission. These findings were consistent with a recent studies conducted in China and United States, where no association was found between co-morbidities and ICU admission [2, 17]. Consistent findings were found in another study conducted in Spain, in which COPD and Asthma were not correlated with ICU admission [3]. Furthermore, another cross-sectional analytical study conducted at a tertiary care center in Nepal similarly found that co-morbidities were not associated with the outcome of COVID-19 [18]. In contrast to our findings, studies conducted in Milan, Italy, and China found that co-morbidities, most commonly hypertension, cardiovascular disease, diabetes, and COPD were related to independent risk factors for death in patients with COVID-19 [19, 20]. Similarly, a study conducted in rural community hospitals in eastern Virginia also revealed that diabetes mellitus and hypertension on admission were predictors of clinical deterioration leading to ICU admission [21]. A meta-analysis done by Jianhua Hu and Yanggan

**Table 3. Severity of COVID-19 at the time of hospital admission.**

| Variables | ICU admission (Cases) n (%) | General admission (control) n (%) | χ2 test p value |
|---|---|---|---|
| Mild COVID—19 | 0 (0.0) | 31 (100.0) | 0.001 |
| Moderate COVID– 19 infection | 0 (0.0) | 7 (100.0) | |
| Moderate COVID– 19 pneumonia | 0 (0.0) | 8 (100.0) | |
| Severe COVID– 19 pneumonia | 33 (62.3) | 20 (37.7) | |

**Table 4. Conditional logistic regression.**

| Variables | β coefficient | COR (95% CI) | AOR (95% CI) | p value |
|---|---|---|---|---|
| **Smoking** | | | | |
| Non- smoker | | Ref | | |
| Smoker | 0.470 | 3.4 (1.4–8.3) | 1.60 (0.77–3.28) | 0.20 |
| **Blood Pressure** | | | | |
| Normal | | Ref | | |
| Elevated | 0.799 | 6.10 (2.40–15.50) | 2.22 (1.05–4.71) | **0.03** |
| **C reactive protein** | | | | |
| Normal | | Ref | | |
| Abnormal | 1.073 | 7.18 (2.77–18.65) | 2.92 (1.24–6.84) | **0.03** |
| **Vaccination** | | | | |
| Vaccinated | | Ref | | |
| Unvaccinated | 0.633 | 6.78 (2.63–17.48) | 1.88 (0.86–4.10) | 0.11 |
| **Preventive practice** | | | | |
| Good | | Ref | | |
| Poor | 1.206 | 12.97 (4.57–36.79) | 3.34 (1.19–9.31) | **0.02** |

Wang also highlighted that the prevalence of co-morbidities such as hypertension, CVD, and diabetes mellitus in severe COVID-19 patients was significantly higher than in non-severe patients [22]. These pathologies have been associated with an increased likelihood of presenting complications and severe forms of the disease. The reported inconsistencies and conflicting findings might be attributed to COVID-19 severity and mortality based on medical conditions specific to geographical regions. Supporting this statement, a systematic review and meta-analysis revealed that the highest mortality among all co-morbidities was observed in European studies compared to Asian studies despite a lower prevalence of co-morbidities [23]. In addition, significant variation in the prevalence of co-morbidities associated with the severity of the disease in different geographical locations was observed [23]. Other potential reasons may include differences in the timing of the study and variations in the spread of COVID-19. The majority of studies were conducted during the early stage of the pandemic, especially in the elderly population. Moreover, differences in variants and sub-variants of the coronavirus might have created discrepancies in the findings. Furthermore, factors such as the sample size of the study, the prevalence of co-morbidities among COVID-19 patients, COVID-19 vaccination rates, diagnostic testing, and the accuracy of co-morbidity might have influenced inconsistent findings. Additionally, the median age of cases and controls in this study was 59 years, which differs from the results of a meta-analysis where median age was over 65 years [22]. These age differences might also have played a role in association with co-morbidities and ICU admission.

Many previous studies found that COVID-19 patients present with lower oxygen saturation, high temperature, pulse rate, respiration rate and blood pressure at the time of hospital admission [17, 18, 20]. These findings are also consistent with the present study. Oxygen saturation is the indicator of respiratory functions and demonstrate variations in arterial oxygenation to indicate hypoxemia in patients. A retrospective study revealed a significantly associated between decreased oxygen saturation and ICU admission [17]. The findings was supported by a nested case- control study conducted in China, wherein patient with SpO2 < 95.5% had 2.35 times higher odds of developing severe COVID-19 [14]. Additionally, a study conducted to identify risk factors for clinical deterioration in patients indicated that reduced oxygen saturation at emergency department was the independent variable associated

with clinical deterioration [3]. A cross sectional comparative study conducted in survivors and non- survivors of Nepal also found that non- survivors tended to exhibit lower oxygen saturation at admission [18].

This study also highlights that elevated blood pressure at the time of hospital admission was independent risk factor in multivariable analysis. Patients presenting with elevated blood pressure at admission were found to be 2.22 times more likely to get admitted to the ICU compared to those with normal blood pressure. This finding was consistent with several studies done in China and Denmark where high systolic blood pressure was found to be significantly associated with treatment in the ICU [2, 24]. Furthermore, a study conducted in Colombia to identify the factors associated with admission to the ICU similarly revealed that an increased mean arterial pressure on admission was significantly associated with patient infected with SARS–CoV- 2 [25].

The pulse rate was found to be significantly associated with ICU admission, a finding consistent with a study conducted in China, where the heart rate beats per minute, was significantly associated with ICU admission [17]. In bivariate analysis, this study found that febrile patients at the time of hospital admission were associated with admission to the ICU. Findings was consistent with study done in China, in which the highest temperature of patients infected with COVID-19 was significantly associated with the requirement of ICU care [2]. Additionally, initial body temperature on admission was found significant predictors of patients requiring intensive unit care [16]. In this study, the respiration rate was also found significantly associated with admission to ICU. The present study showed that patient presenting with tachypnea during the time of hospital admission were 15.5 times more likely to be admitted in ICU. These findings was alike with several studies where the respiration rate was found to be significantly associated with ICU admission [3, 14, 17].

Higher leucocyte count typically indicates that the body is fighting infection or inflammation. In this study, the total leucocyte count at the time of hospital admission was found to be associated with admission to the ICU. Findings are consistent with study conducted in Italy where white blood cells count was significantly associated with the risk of death [19]. Similar, multicenter retrospective case- control study, where WBC was significantly associated with non–survivor group [20]. Furthermore, clinical mortality review conducted in Sukraraj Tropical and Infectious Disease Hospital, among 50 mortalities reviewed 80% of patients had lymphocytopenia [26]. However, a contrast finding was concluded by a study done in China where white blood cells were not significantly associated with the severity of COVID-19 [14].

C- reactive protein is a protein made by the liver and serve as a marker for acute inflammation. In present study, CRP was found to be significant predictors of ICU admission. This findings was similar to study conducted in China where CRP was found to be an independent predictors associated with severe COVID-19 [14]. Another study conducted in China also found CRP significantly associated with ICU admission [17]. Recent meta-analysis also revealed that most common blood test abnormalities were elevated C- reactive protein levels [22].

The present study revealed no significant association between COVID-19 vaccination status, whether with the first or second dose of any available vaccine, and ICU admission in multivariate analysis. These findings were consistent with a study conducted at the central hospital of Nepal, which similarly showed no significant association between vaccination status and the outcome of COVID-19 [18]. In contrast to these findings, a study conducted in England reported vaccination was associated with a significant reduction in symptomatic SARS–CoV2 positive cases [27]. The discrepancy in findings might be due to variations in natural immunity among people of different countries.

Present study concluded that patients admitted to the ICU had a longer hospital stay compared to patients admitted in COVID-19 isolation ward, which is indicating a significant association. In this study, median hospital stay among cases was 13 days while it was 2 days among controls. This findings was consistent with a study conducted in Wuhan where the median time from onset of symptoms to ICU admission was 10.5 days [2]. One more study conducted in China showed that the total in hospital days were 20 days among survivors and 15 days among non–survivor [20]. Additionally another study conducted in Nepal, showed median ICU stay days to be 10 days in the survivors group and 8 days in non–survivors group [18]. The possible explanation could be a longer duration of stay in ICU due to disease severity, leading to a longer phase of recovery.

Present study found that application of preventive measures was found to be independently associated with ICU admission. Similar findings were observed in a systematic review and meta-analysis, where non pharmaceutical Public health interventions showed 16.5% decrease in ICU admission [28]. Similarly, another study conducted in Thailand also showed practices such as wearing masks, handwashing and social distancing were independently associated with a lower risk for SARS–CoV-2 infection [29]. A systematic review and meta- analysis revealed that several personal protective and social measures such as physical distancing, hand washing and wearing masks were associated with reductions in the incidence of COVID -19 [30, 31].

This study is of retrospective nature, participants may fail to remember the preventive measures they applied before two weeks of being infected with COVID -19. Thus, it may result in recall bias. The present study was carried out with small sample size based on a single institution. Therefore, the results of the study can only be generalized to patients within the Sukraraj Tropical and Infectious Disease Hospital.

## Conclusion

The present study presents findings suggesting that elevated blood pressure, abnormal C-reactive protein levels, and inadequate preventive practices serve as predictors and risk factors associated with ICU admission. Furthermore, this investigation delineates specific clinical features and laboratory abnormalities observed at the time of hospital admission that contribute to heightened risk.

Assessment of these factors upon admission holds the potential for facilitating risk stratification and early triage of individuals at risk of severe COVID-19 illness, thereby enabling timely and optimal management strategies for COVID-19 cases. Additionally, policymakers are urged to prioritize the enhancement of public health measures during pandemics of a similar nature.

## Supporting information

**S1 Text. Questionnaire to assess prevention practices against COVID– 19 in the general population.**
(DOCX)

**S2 Text. Nepali version of preventive practice questionnaire.**
(DOCX)

## Acknowledgments

We would like to acknowledge all participants who allocated time to our study and all people who directly and indirectly supported our study.

## Author Contributions

**Conceptualization:** Dipsikha Aryal, Paras Kumar Pokharel.

**Data curation:** Vijay Kumar Khanal, Gyanu Nepal Gurung, Sudikshya Neupane, Shikha Basnet.

**Formal analysis:** Dipsikha Aryal, Vijay Kumar Khanal, Gyanu Nepal Gurung, Sudikshya Neupane, Shikha Basnet.

**Investigation:** Bimal Sharma Chalise.

**Methodology:** Dipsikha Aryal, Paras Kumar Pokharel, Vijay Kumar Khanal, Gyanu Nepal Gurung, Bimal Sharma Chalise, Sudikshya Neupane, Shikha Basnet.

**Project administration:** Anup Ghimire, Shikha Basnet.

**Resources:** Dipsikha Aryal, Bimal Sharma Chalise, Shikha Basnet.

**Software:** Vijay Kumar Khanal, Gyanu Nepal Gurung, Sudikshya Neupane.

**Supervision:** Dipsikha Aryal, Paras Kumar Pokharel, Anup Ghimire, Gyanu Nepal Gurung, Bimal Sharma Chalise.

**Validation:** Gyanu Nepal Gurung, Bimal Sharma Chalise.

**Visualization:** Paras Kumar Pokharel, Bimal Sharma Chalise.

**Writing – original draft:** Dipsikha Aryal, Gyanu Nepal Gurung, Sudikshya Neupane.

**Writing – review & editing:** Paras Kumar Pokharel, Anup Ghimire, Vijay Kumar Khanal, Shikha Basnet.

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
