## [Decision Letter · Decision Letter 0]

16 Aug 2023

PGPH-D-23-00924

PREDICTORS TO INTENSIVE CARE UNIT ADMISSION AMONG PATIENT WITH CORONAVIRUS DISEASE IN SUKRARAJ TROPICAL AND INFECTIOUS DISEASE HOSPITAL, NEPAL: A CASE CONTROL STUDY

Dear Dr. Aryal,

Thank you for submitting your manuscript to PLOS Global Public Health. After careful consideration, we feel that it has merit but does not fully meet PLOS Global Public Health’s publication criteria as it currently stands. Therefore, we invite you to submit a revised version of the manuscript that addresses the points raised during the review process.

We look forward to receiving your revised manuscript.

Kind regards,

André Machado Siqueira, M.D., MSc, Ph.D

Academic Editor

Journal Requirements:

1. In the ethics statement in the Methods, you have specified that verbal consent was obtained. Please provide additional details regarding how this consent was documented and witnessed, and state whether this was approved by the IRB.

2. Please provide separate figure files in .tif or .eps format only and remove any figures embedded in your manuscript file. Please also ensure all files are under our size limit of 10MB.

4. Please amend your Data Availability Statement and indicate where the data may be found.

Additional Editor Comments (if provided):

Reviewers' comments:

Reviewer's Responses to Questions

**Comments to the Author**

1. Does this manuscript meet PLOS Global Public Health’s publication criteria? Is the manuscript technically sound, and do the data support the conclusions? The manuscript must describe methodologically and ethically rigorous research with conclusions that are appropriately drawn based on the data presented.

Reviewer #1: Partly

Reviewer #2: No

2. Has the statistical analysis been performed appropriately and rigorously?

Reviewer #1: No

Reviewer #2: No

3. Have the authors made all data underlying the findings in their manuscript fully available (please refer to the Data Availability Statement at the start of the manuscript PDF file)?

Reviewer #1: Yes

Reviewer #2: No

4. Is the manuscript presented in an intelligible fashion and written in standard English?

Reviewer #1: Yes

Reviewer #2: Yes

5. Review Comments to the Author

Reviewer #1: Dear authors;

I applaud your work on preparing this paper. Unfortunately, I have some concerns that I would like to see addressed to improve the work and ensure the conclusions are valid.

Minor:

1. Methods section needs restructuring.

Would benefit from a list of inclusions and exclusions early on, to include age restrictions and outcome (in-hospital death).

Sample selection: Line 193 From my understanding, the 99 patients were not randomly selected. Rather, they identified ICU admissions with COVID-19 (who survived discharge) and then matched them to other patients with admitted to the ward with COVID-19. Lines 109-110: I think they are saying every 2nd or every 3rd. It is probably every 3rd patient was reviewed. Lines 114-115. They mention “inclusion criteria”, but I do not see where those are listed.

I think they are trying to say they selected 2 controls based on each case.

Analysis: Lines 242-245 belong in the methods section ”variables that were significantly associated with ICU admission, with a p-value 243 less than or equal to 0.2 in bivariate analysis were taken for multivariate logistics regression 244 analysis. Variables with expected count less than 5 were not included in multivariate logistics 245 regression analysis.”

2. Lines 120-126 these are results (as per Figure 1).

3. Figure 1- it is surprising that all deaths occurred in ward patients. They should check that. Also, why are the ICU patients separated into the ventilator group? Were those cases ineligible for the study as well?

Main concerns are:

1. Recall bias – Data collection occurred in June – July 2022. The population contacted were in hospital between July 2021 and Jan 2022. This suggests telephone interviews asking about symptoms and preventative measures in the 2 weeks prior to their hospitalisation occurred between six months and a year after their hospitalisation. Preventative measures recall may be worse and/or biased for ICU patients compared to ward patients. Some variables are likely okay to collect six months after hospitalisation– e.g. smoking, comorbidities and vaccination status. I would not conclude anything, however, about other variables due to bias.

2. Data analysis: The analysis they did was flawed, hence I cannot yet believe the findings. Logistic regression model likely needed to be conditional due to the matching procedure. Inclusion of the matching variables (age and sex) is required at a minimum for matched studies. How was inter-dependence of variables assessed? Eg. Smoking and preventative practices? Smoking and sex? Vaccination status and smoking? With a sample of only 100 in total, no more than 5 or 6 covariates should be included. Sex and age need to be included, then they can try and build a parsimonious model with other variables with a large B-coefficient, after assessing for interdependence/ intercorrelations. Did they test COPD in multivariable model? And COPD was likely associated with smoking. Hence they need to include the interaction term of COPD*smoking or pick the better predictor of the two. The Nagelkerke R2 is very high. I would be surprised if the variables in the model explained that much of the variance. Furthermore, duration of hospital stay isn’t a predictor of ICU admission, it’s likely a result of ICU admission. That should not be included.

Lastly, They discuss “severity at admission” in the methods section but do not present that data anywhere in the tables. It would be interesting to see that data, especially if the ward patients were the ones that were dying. Did patients with a “do not resuscitate” plan go to the ward directly? Age categories would also be helpful to see, e.g. 18-39 years, 40-59 years, 60-79 years and 80+ years.

Your discussion is engaging and interesting. There are some grammatical errors throughout the paper, but overall very well written. If the paper will be published in English, then some language editing would be recommended.

Thank you for the opportunity to review this paper.

Reviewer #2: **Major Changes:**

Minimum data was not provided, therefore it is impossible to verify the results.

Study site and population not properly described.

The flowchart contains errors, as the numbers do not add up.

The flowchart follows a cohort format, which is an inappropriate study design.

It is recommended to approach the work as a retrospective cohort design or a cross-sectional analysis instead of a case-control study.

What was the power of the study calculated? The sample size (N) appears too small.

The study sample size was small; consequently, the statistical results yielded extremely wide confidence intervals. While these intervals allow us to determine whether a variable is associated with the model, they do not facilitate additional interpretations that contribute to understanding the risk probability.

The abstract mentions "The adjusted multivariable logistics regression," but this information is not thoroughly explained in the methods section. Furthermore, the "multivariable logistics regression" only appears in Table 5, where it is referred to as 'binary logistic regression'. It is unclear whether the authors used binary or multiple logistic regression.

Report the number of instances of "missing data" for each variable.

Include both crude and adjusted odds in the same Table 5.

The reference levels for each variable in the logistic regression (Tables 3 & 4) are not clearly specified.

For the variable "COVID-19 vaccination" (Yes/No), the reference level should be set as "Yes." This adjustment allows the "No" value to indicate the significantly increased risk for non-vaccinated individuals being hospitalized in the ICU.

Line 371: The sentence "result may be generalized only to patient under similar setting" is incorrect. The study's results cannot be generalized to patients under similar settings. The results of the study can only be generalized to patients within the same hospital.

**Minor Changes:**

Table 4 is incorrectly numbered.

In Table 5, provide the number of cases and controls for each variable.

Overall, the text appears to be well-written, but there are a few areas where improvements can be made.

Line 51: Change "pandemic in March 2020(1)" to "pandemic in March 2020 (1)."

Line 55: Change "major symptoms like fever, cough, dyspnea" to "major symptoms such as fever, cough, and dyspnea."

Line 56: Change "cutaneous manifestations are common clinical features of this disease (2)" to "cutaneous manifestations are common clinical features of this disease (2)."

Line 64: Change "The first case of coronavirus was seen in a Nepalese Citizen returning" to "The first case of coronavirus was observed in a Nepalese citizen returning."

Line 69: Change "endorsed in Nepal from 25th March 2020 (6)" to "endorsed nationwide in Nepal on March 25, 2020 (6)."

Line 104: Change "public hospital with High Dependency Unit" to "public hospital with a High Dependency Unit."

Line 242: Change "variables that were significantly associated with ICU admission" to "variables significantly associated with ICU admission."

Line 259: Change "COVID-19 infected patients was found to be 31.13%" to "The proportion of COVID-19 infected patients was found to be 31.13%."

6. PLOS authors have the option to publish the peer review history of their article (what does this mean?). If published, this will include your full peer review and any attached files.

**Do you want your identity to be public for this peer review?** For information about this choice, including consent withdrawal, please see our Privacy Policy.

Reviewer #1: **Yes: **Amy Lynn Sweeny

Reviewer #2: No

---

## [Decision Letter · Decision Letter 1]

7 Dec 2023

PGPH-D-23-00924R1

PREDICTORS TO INTENSIVE CARE UNIT ADMISSION AMONG PATIENT WITH CORONAVIRUS DISEASE IN SUKRARAJ TROPICAL AND INFECTIOUS DISEASE HOSPITAL, NEPAL: A CASE CONTROL STUDY

Dear Dr. Aryal,

Thank you for submitting your manuscript to PLOS Global Public Health. After careful consideration, we feel that it has merit but does not fully meet PLOS Global Public Health’s publication criteria as it currently stands. Therefore, we invite you to submit a revised version of the manuscript that addresses the points raised during the review process.

We look forward to receiving your revised manuscript.

Kind regards,

André Machado Siqueira, M.D., MSc, Ph.D

Academic Editor

Journal Requirements:

1. In the ethics statement in the Methods, you have specified that verbal consent was obtained. Please provide additional details regarding how this consent was documented and witnessed, and state whether this was approved by the IRB.

Additional Editor Comments (if provided):

Reviewers' comments:

Reviewer's Responses to Questions

**Comments to the Author**

1. If the authors have adequately addressed your comments raised in a previous round of review and you feel that this manuscript is now acceptable for publication, you may indicate that here to bypass the “Comments to the Author” section, enter your conflict of interest statement in the “Confidential to Editor” section, and submit your "Accept" recommendation.

Reviewer #1: (No Response)

Reviewer #2: All comments have been addressed

2. Does this manuscript meet PLOS Global Public Health’s publication criteria? Is the manuscript technically sound, and do the data support the conclusions? The manuscript must describe methodologically and ethically rigorous research with conclusions that are appropriately drawn based on the data presented.

Reviewer #1: Partly

Reviewer #2: Yes

3. Has the statistical analysis been performed appropriately and rigorously?

Reviewer #1: No

Reviewer #2: I don't know

4. Have the authors made all data underlying the findings in their manuscript fully available (please refer to the Data Availability Statement at the start of the manuscript PDF file)?

Reviewer #1: Yes

Reviewer #2: Yes

5. Is the manuscript presented in an intelligible fashion and written in standard English?

Reviewer #1: Yes

Reviewer #2: No

6. Review Comments to the Author

Reviewer #1: The authors have improved the study in its revised version, but I still have concerns about the statistical methods used and conclusions stated in this study.

The study’s sample size calculation doesn’t make sense. The “disease” is ICU admission in this study, the “exposure” is any of: smoking, high bp, etc. To do the sample size you would pick one of these (whichever is the most important in your opinion) and have an estimate of how common that exposure one in the cases and the controls. Then you could get a valid sample size.

Since the study didn’t do sample size calculation in this manner, I recommend deleting the sample size calculation paragraphs.

The period of review was 16th July 2021- 14th January 2022 according to page 6 line 108.

The authors excluded patients who died. Please explain why this is justified? If it is because you can’t ascertain the “preventative measures”, again, this is a flaw in the study design. Any findings of the study can be interpreted as valid only for people who survive COVID-19.

I appreciate the authors use of conditional logistic regression modelling for matched data. However, the authors also have to include the matching variables – age and sex- in the model for the model to be accurate. The results for age and sex don’t matter, but they may effect the other exposures of interest and need to be included in the model.

In results, the authors state that 99 patients were randomly selected from the population of PCR+ patients. This is not the case. 33 patients admitted to ICU who were able to be contacted post discharge were selected randomly from all patients admitted to ICU and able to be contacted post-discharge, then these were matched 2:1 to other patients admitted to the general ward who were able to be contacted post discharge.

Tables 2 to 5 present crude odds ratios. This should be qualified as such in the titles. Also, whether or not it is valid to present crude odds ratios for a matched study is debatable. In tables 2-5, the authors should consider presenting a) crude odds ratios and b)AOR for the variable, adjusting for age and sex.

Reviewer #2: Congratulations on the manuscript improvement. However, some points still need adjustments and clarification.

**Major:**

1. English requires revision. Several phrases lack proper punctuation, making it difficult to read. I have pointed out some adjustments, but only a few. The English is well-written and mostly easy to read, but an extensive grammar review is required.

2. The flowchart still needs adjustments (check comments in the PDF-reviewed file). I wonder if it is necessary to be in the text.

3. Tables need adjusting. Tables 2 to 5 can be merged.

4. Check manuscript formatting at [link1] and [link2].

5. In the bivariate results, it is not clear why only blood pressure, C-reactive protein, and COVID-19 vaccination variables from Table 4 were included in the conditional logistic regression?

**Minor:**

1. Standardize the use of 'full-written-words' and abbreviations.

2. Add an additional file with maps/geographical visualization of the country/state/province and hospital location. That will help foreign readers better understand the study's epidemiological aspects. I suggest using the following model: [link3].

3. Table 6 is not in a good format; I suggest using the following model: [link4].

4. There is room for improvement in the last paragraph of the discussions section on the matter of 'study generalization,' and in the conclusions section, provide more direct conclusions on how to use the results found to improve public health actions/strategies/policies, although it is already implicit in discussions.

7. PLOS authors have the option to publish the peer review history of their article (what does this mean?). If published, this will include your full peer review and any attached files.

**Do you want your identity to be public for this peer review?** For information about this choice, including consent withdrawal, please see our Privacy Policy.

Reviewer #1: No

Reviewer #2: No

---

## [Decision Letter · Decision Letter 2]

20 Feb 2024

PGPH-D-23-00924R2

PREDICTORS TO INTENSIVE CARE UNIT ADMISSION AMONG PATIENT WITH CORONAVIRUS DISEASE IN SUKRARAJ TROPICAL AND INFECTIOUS DISEASE HOSPITAL, NEPAL: A CASE CONTROL STUDY

Dear Dr. Aryal,

Thank you for submitting your manuscript to PLOS Global Public Health. After careful consideration, we feel that it has merit but does not fully meet PLOS Global Public Health’s publication criteria as it currently stands. Therefore, we invite you to submit a revised version of the manuscript that addresses the points raised during the review process.

We look forward to receiving your revised manuscript.

Kind regards,

Reuben Kiggundu

Academic Editor

Journal Requirements:

1. In the ethics statement in the Methods, you have specified that verbal consent was obtained. Please provide additional details regarding how this consent was documented and witnessed, and state whether this was approved by the IRB"

Additional Editor Comments (if provided):

Reviewers' comments:

Reviewer's Responses to Questions

**Comments to the Author**

1. If the authors have adequately addressed your comments raised in a previous round of review and you feel that this manuscript is now acceptable for publication, you may indicate that here to bypass the “Comments to the Author” section, enter your conflict of interest statement in the “Confidential to Editor” section, and submit your "Accept" recommendation.

Reviewer #2: All comments have been addressed

Reviewer #3: (No Response)

2. Does this manuscript meet PLOS Global Public Health’s publication criteria? Is the manuscript technically sound, and do the data support the conclusions? The manuscript must describe methodologically and ethically rigorous research with conclusions that are appropriately drawn based on the data presented.

Reviewer #2: Yes

Reviewer #3: Partly

3. Has the statistical analysis been performed appropriately and rigorously?

Reviewer #2: Yes

Reviewer #3: Yes

4. Have the authors made all data underlying the findings in their manuscript fully available (please refer to the Data Availability Statement at the start of the manuscript PDF file)?

Reviewer #2: Yes

Reviewer #3: Yes

5. Is the manuscript presented in an intelligible fashion and written in standard English?

Reviewer #2: Yes

Reviewer #3: No

6. Review Comments to the Author

Reviewer #2: I am satisfied with the corrections made. However, after reading the comments from reviewer 1, I am uncertain whether the description of the sampling calculation should indeed be included in the main text.

The flowchart still requires improvements as it lacks information, and the mathematics do not align. Please check the additional reviewer file to see the infor that are missing. The repositioning of the 4th box should be under box 3. Additionally, the flowchart is not visually pleasing, and the arrows are not aligned. Kindly adjust the flowchart again or consider removing it.

The map figure lacks subtitles. What do the red GPS markers represent? And what about the blue ones? What do the blue-painted zones indicate? The map also lacks a scale and a compass rose. I suggest inserting a small figure of the country, indicating where the study is conducted. Please refer to the example in the file I sent.

Reviewer #3: 1. The main issue this article is dealing with is widely studied and a significant number of articles about it are published so far. The fact that in Nepal this might be the first study of its kind but this is not relevant enough for publishing.

2. The Introduction section should inform the reader about the knowledge that we already have about the research subject, and after that to explain what were the reasons that this study was conducted. The Patients and Methods section has to much better organized and written. Authors mention inclusion criteria, and those criteria are not explained; what was the reason to exclude 105 COVID-19 patients that died in non-ICU wards?; authors mention the stratification of patients according to the disease severity, yet it is not present in the text later; how can the length of stay in the hospital be used as a predictor for the ICU admission?; the methodology for the questionnaires use has to be properly explained; in this study high blood pressure at the admission was a predictor for the ICU admission, yet no comorbidities, including hypertension was not a predictor for the ICU admission...? The Discussion section has to be better organized and much better written, avoiding speculations like the one when authors try to explain the differences in natural immunity between people of different countries. The Conclusions section should be re-written after the manuscript revision.

3. The report is written in rather poor English language and it is highly recommended that the manuscript should be revised by a native English language speaker.

7. PLOS authors have the option to publish the peer review history of their article (what does this mean?). If published, this will include your full peer review and any attached files.

**Do you want your identity to be public for this peer review?** For information about this choice, including consent withdrawal, please see our Privacy Policy.

Reviewer #2: No

Reviewer #3: No

---

## [Editor Report · Decision Letter 3]

29 Feb 2024

PREDICTORS TO INTENSIVE CARE UNIT ADMISSION AMONG PATIENT WITH CORONAVIRUS DISEASE IN SUKRARAJ TROPICAL AND INFECTIOUS DISEASE HOSPITAL, NEPAL: A CASE CONTROL STUDY

PGPH-D-23-00924R3

Dear Garcia,

We are pleased to inform you that your manuscript 'PREDICTORS TO INTENSIVE CARE UNIT ADMISSION AMONG PATIENT WITH CORONAVIRUS DISEASE IN SUKRARAJ TROPICAL AND INFECTIOUS DISEASE HOSPITAL, NEPAL: A CASE CONTROL STUDY' has been provisionally accepted for publication in PLOS Global Public Health.

Best regards,

Reuben Kiggundu

Academic Editor
